# Oxidation Behavior of (Mo,Hf)Si_2_-Al_2_O_3_ Coating on Mo-Based Alloy at Elevated Temperature

**DOI:** 10.3390/ma16083215

**Published:** 2023-04-19

**Authors:** Yongqi Lv, Huichao Cheng, Zhanji Geng, Wei Li

**Affiliations:** State Key Laboratory of Powder Metallurgy, Central South University, Changsha 410083, China; yqlv2008@csu.edu.cn (Y.L.);

**Keywords:** (Mo,Hf)Si_2_-Al_2_O_3_ composite coating, multi-layer structure, oxidation behavior, phase transformation, microstructure evolution

## Abstract

To improve the oxidation resistance of Mo-based alloys, a novel (Mo,Hf)Si_2_-Al_2_O_3_ composite coating was fabricated on a Mo-based alloy by the method of slurry sintering. The isothermal oxidation behavior of the coating was evaluated at 1400 °C. The microstructure evolution and phase composition of the coating before and after oxidation exposure were characterized. The anti-oxidant mechanism for the good performance of the composite coating during high-temperature oxidation was discussed. The coating had a double-layer structure consisting of a MoSi_2_ inner layer and a (Mo,Hf)Si_2_-Al_2_O_3_ outer composite layer. The composite coating could offer more than 40 h of oxidation-resistant protection at 1400 °C for the Mo-based alloy, and the final weight gain rate was only 6.03 mg/cm^2^ after oxidation. A SiO_2_-based oxide scale embedded with Al_2_O_3_, HfO_2_, mullite, and HfSiO_4_ was formed on the surface of the composite coating during oxidation. The composite oxide scale exhibited high thermal stability, low oxygen permeability, and enhanced thermal mismatch between oxide and coating layers, thus improving the oxidation resistance of the coating.

## 1. Introduction

Molybdenum (Mo) alloys are of great interest as high-temperature structural materials because of their attractive properties, such as high melting points, outstanding mechanical properties at elevated temperatures, low coefficient of thermal expansion (CTE), high thermal conductivity, and good thermal shock resistance [1,2,3,4,5]. In practical applications, high-temperature structural materials are not only required to have excellent high-temperature mechanical properties, but also withstand complex environments, such as high-temperature oxidation, corrosion, and ablation. Nevertheless, the catastrophic behavior of molybdenum alloys under high-temperature oxidizing environments has been a key bottleneck to their high-temperature applications [6,7]. An oxidation-resistant coating is necessary for the high-temperature applications of these Mo-based alloys in oxidizing environments.

Silicide is an intermetallic material that has excellent oxidation resistance at high temperatures resulting from the formation of a continuous and protective SiO_2_ layer at temperatures above 1000 °C, and has been widely used as an oxidation-resistant coating for refractory metals [8,9,10,11,12]. Among them, MoSi_2_ has attracted much attention due to its high melting point (2030 °C), moderate density of 6.24 g/cm^3^, and similar CTE (8.1 × 10^−6^/°C) to refractory metals (e.g., Mo 6.7 × 10^−6^/°C). However, MoSi_2_ suffers from rapid pesting oxidation between 400 and 600 °Cm which can lead to the disintegration of the material [13]. In addition, it performs limited protective performance at higher temperatures due to the volatilization of SiO_2_, which has hindered its use under high-temperature oxidizing environments [14,15]. Attempts have been made to introduce alloying elements such as aluminum (Al), titanium (Ti), zirconium (Zr), and hafnium (Hf), as well as their oxides (e.g., Al_2_O_3_, ZrO_2_, HfO_2_), into MoSi_2_-based coatings to enhance their properties at elevated temperatures [16,17,18,19,20,21,22]. Hf has been widely adopted to improve the high-temperature performance of superalloys [23,24,25,26] (e.g., Ni-, Co-, Fe-, and Nb-based superalloys) and the high-temperature coatings on those superalloy substrates (e.g., β-NiAl coating with B2 structure [27], aluminide coating [28], AlCoCrFeNi high-entropy alloy coating [29]). These reports have shown that small additions of Hf can effectively increase the oxide scale adhesion and thermal stability, thus increasing the service life of the coating [30]. There have been discussions on the effect of Hf on MoSi_2_-based composites and coatings. Refs. [31,32] declared that Mo-Si-B bulk material doping with Hf exhibited enhanced mechanical properties and oxidation resistance at 1650 °C due to the formation of dense HfO_2_ and HfSiO_4_ oxides. Refs. [18,33] reported that the Hf-doped Mo-Si-B coating exhibits excellent oxidation resistance due to the formation of a protective surface oxide layer based on SiO_2_+HfO_2_+HfSiO_4_. In addition, Al_2_O_3_ has been a favorable additive for MoSi_2_ due to its high melting point (2054 °C), high thermal stability, and moderate CTE (8.3 × 10^−6^/°C) similar to that of MoSi_2_ (8.1 × 10^−6^/°C). It has been confirmed that the addition of Al_2_O_3_ to MoSi_2_ coatings is beneficial to restrain pest oxidation at low temperatures (500 °C) and improve high-temperature oxidation resistance (above 1500 °C) by generating a SiO_2_-mullite composite oxide layer with high thermal stability and low oxygen permeability [5,17,34]. In recent years, attempts to synthesize MoSi_2_ composites with superior properties for high-temperature applications by combining the alloying element (e.g., Ti, Al) and composite (e.g., Al_2_O_3_) have been reported [16,35]. Zhang C. et al. [16] reported the preparation and corrosion behavior of a Al and Al_2_O_3_ co-doped MoSi_2_-based coating on the surface of molybdenum. Tian et al. [36] investigated the effect of Y_2_O_3_/yttrium (Y) on the Mo-Si-B coating on pure Mo.

In this work, Hf and Al_2_O_3_ were co-doped to a MoSi_2_ coating to form a new (Mo,Hf)Si_2_-Al_2_O_3_ composite coating on a Mo-based alloy substrate. Among the various coating techniques to prepare MoSi_2_-based silicide coatings on refractory metals, vacuum slurry sintering is probably the most competitive, which offers the advantages of being low-cost, effective, and easy to operate [37,38]. Moreover, a metallurgical bond between the coating and substrate can be expected. Accordingly, the Hf-Al_2_O_3_ co-doped MoSi_2_ coating was fabricated on a Mo-based substrate by the method of slurry sintering. The isothermal oxidation behavior of the composite coating was evaluated at 1400 °C in static air. The microstructure evolution, element distribution, and phase composition of the coatings before and after oxidation were characterized, and the antioxidant mechanism of the coating at high temperatures was also discussed.

## 2. Materials and Methods

### 2.1. Sample Preparation

Long strip specimens (80 mm × 10 mm × 2 mm) used as substrates were cut from Mo-0.7 wt.% La_2_O_3_ alloy plate. All samples were hand-polished with SiC-grit papers of 400, 600, and 1000 mesh, cleaned in an ultrasonic ethanol bath, and then dried in vacuum at 80 °C for 2 h.

The (Mo,Hf)Si_2_-Al_2_O_3_ coating was prepared on Mo-La_2_O_3_ alloy substrates through the slurry sintering process. Firstly, a Si-Mo-Hf-Al_2_O_3_ slurry was prepared from a mixture of pure Si powders (≥99%, 1~3 μm), Mo powders (≥99%, 1~2 μm), Hf powders (>98%, 1~2 μm), and Al_2_O_3_ powders (≥99%, 5~10 μm) with a weight ratio of 65:20:5:10 for Si:Mo:Hf: Al_2_O_3_. A small amount of halide (NH_4_F) and organic binder (nitrocellulose) was added to the powder mixture, and then the powder mixture was attrition milled for 6 h using ethyl acetate as solvent. Subsequently, the slurry was sprayed evenly on the surface of the Mo-La_2_O_3_ alloy substrate through an air compressor and a manual spray gun until a suitable coating thickness was obtained. The spraying process parameters for the Si-Mo-based slurry have been reported in previous work [5]. Finally, the as-sprayed samples were dried using an infrared baking lamp, taken into a vacuum furnace, and then sintered at 1480 °C for 20 min in vacuum (<1 Pa). Subsequently, the sintered coating samples were furnace-cooled to room temperature.

### 2.2. Oxidation Test and Characterization

Isothermal oxidation tests were conducted at 1400 °C in static air using an electric furnace, equipped with real-time display and automatic data recording. The sample temperature was measured by infrared radiation thermometer. Mass change of the specimens after oxidation for different times was measured using an analytical balance with an accuracy of 10^−4^ g to study the oxidation characteristics of the composite coating. The weight gain P_0_ can be calculated by the following equation:P_0_ = △m/S(1)
where △m is the mass gain of the coating after oxidation (mg), and S refers to the superficial area of coatings (cm^2^).

The phase composition of the coating before and after oxidation was analyzed by X-ray diffraction (XRD, D/Max 2500, Cu-Ka radiation, Tokyo, Japan). The surface and cross-sectional morphology of the as-coated and oxidized coating were obtained using scanning electron microscopy (SEM, FEI Sirion 200, Waltham, MA, USA) coupled with an energy-dispersive spectrometer (EDS) device. An electron-probe micro-analyzer (EPMA, JEOL JXA-8230, Musashino, Japan) was utilized to characterize the elemental distribution of the coating before and after oxidation.

## 3. Results and Discussion

### 3.1. Microstructure and Phase Composition of the Coating

Figure 1a shows the back-scattered electronic (BSE) image of the surface of the as-coated sample. The coating surface was relatively loose and rough, and mainly consisted of two kinds of particles. Meanwhile, there were some pores and pits on the coating surface; however, no noticeable microcrack was observed. It can be seen that dark gray particles (Spot 1) and white particles (Spot 2) were embedded with each other on the coating surface, while a number of white particles became intermingled together to form island-like clusters, as indicated by the blue arrows. The particle size of dark gray particles was in the range of 5~10 μm. The maximum size of the white island-like clusters was up to 20 μm while the minimum size of small particles was less than 2 μm. EDS results in Table 1 reveal that the composition of Spot 1 contained Al and O elements, and the ratio of Al:O was approximately 2:3, while the composition of Spot 2 contained Mo, Hf, and Si elements, and the ratio of (Mo+Hf):Si was about 1:2. Figure 1b–f shows the element mapping of the surface of the coating, which further confirms the distribution and element composition of the white particles and dark gray particles. Al and O elements were mainly distributed in the dark gray particles (Spot 1 in Figure 1a, while Mo and Si elements were mainly distributed in the white particles (Spot 2 in Figure 1a). It is noteworthy that the distribution of Hf was similar to that of Mo and Si, which was also distributed mainly in the white particles. 

Figure 2 displays the XRD pattern of the surface of the as-prepared coating. According to the XRD pattern, the coating was composed of MoSi_2_ and Al_2_O_3_. No other phases were detected. Combined with the XRD, EDS, and element mapping results, dark gray particles (Spot 1) and white particles (Spot 2) could be identified to be MoSi_2_ with Hf existing in the solid solution and Al_2_O_3_, respectively. 

**Figure 1 materials-16-03215-f001:**
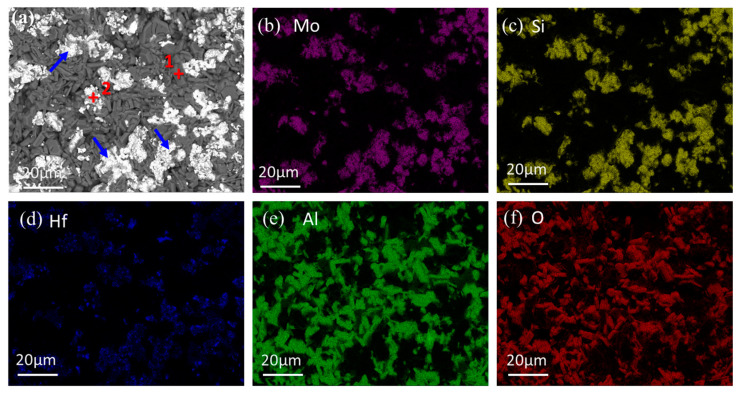
(**a**) Surface morphology, (**b**–**f**) element mapping of the surface of the as-prepared coating.

**Table 1 materials-16-03215-t001:** EDS analysis of Spots 1–2 in Figure 1 and Spots 3–5 in Figure 3.

Spot	Composition (at.%)					Main Phase
	Mo	Hf	Si	Al	O	
1	0.29	0.05	0.61	39.42	59.63	Al_2_O_3_
2	30.83	2.09	63.06	1.17	2.85	(Mo,Hf)Si_2_
3	31.55	1.62	64.43	0.83	1.57	(Mo,Hf)Si_2_
4	32.32	-	67.25	0.15	0.28	MoSi_2_
5	0.51	0.04	0.84	38.90	59.71	Al_2_O_3_

**Figure 2 materials-16-03215-f002:**
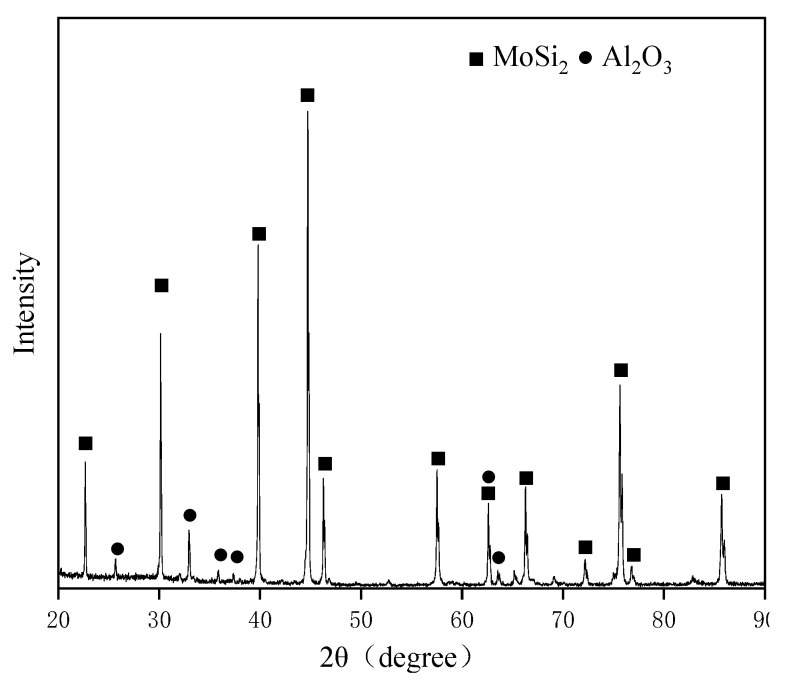
XRD pattern of the surface of the as-prepared coating.

**Figure 3 materials-16-03215-f003:**
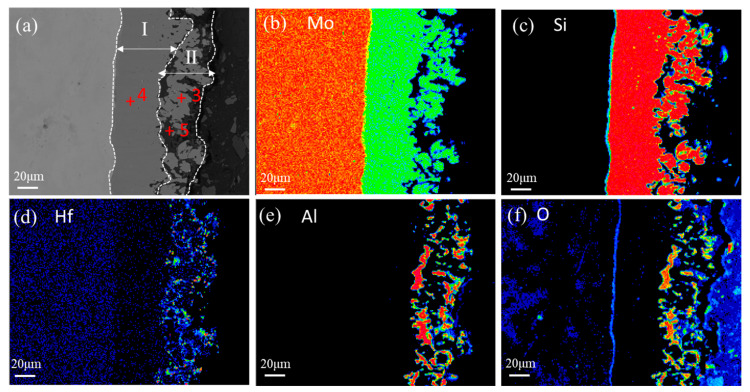
(**a**) Cross-sectional morphology; (**b**–**f**) element mapping of the as-coated coating.

Figure 3a shows the cross-sectional microstructure of the as-prepared coating. A typical double-layer structure consisting of an outer layer (Layer I) and an inner layer (Layer II) can be observed in the coating. There was no obvious diffusion layer between the coating and the substrate. Apparently, the inner layer showing a dense structure was firmly adhered to the Mo-based alloy substrate and constituted the principal part of the coating, which had 48~50 μm of thickness, while the outer layer showed a loose but continuous structure, which had 30~38 μm of thickness. The outer layer also adhered well to the inner layer, which is an important requirement for the long anti-oxidation life of the coating. Gray and black gray phases were observed inlaid with each other in the outer layer. Figure 3b–f displays the element mapping of the cross-section of the coating in Figure 3a. As revealed by the element mapping, the inner layer was rich in Mo and Si elements, while the outer layer was rich in Mo, Si, Hf, Al, and O. It seems Hf mainly distributed uniformly in the Mo-Si rich phases in the outer layer, and barely any of the Hf element appeared in the inner layer. Element mapping combined with EDS analysis and XRD pattern reveals the phase distribution, wherein Spots 3 and 4 were (Mo,Hf)Si_2_ and MoSi_2_ phases, respectively, while the black particles (Spot 5) were Al_2_O_3_. It can be inferred that the inner layer was mainly composed of MoSi_2_, while the outer part of the coating was identified to be a (Mo,Hf)Si_2_-Al_2_O_3_ mixed layer.

Figure 4 shows a schematic diagram of the microstructure evolution of the composite coating during preparation. The slurry containing Si, Mo, Hf, and Al_2_O_3_ powders was sprayed onto the surface of the Mo-based alloy substrate and then sintered at 1480 °C. Since the sintering temperature was above its melting point (1410 °C), Si was in a molten state and diffused inward to the direction of the matrix and reacted with the Mo-based substrate to form a Mo-silicide inner layer by liquid phase reaction, resulting in a relatively dense structure. High melting point elements Mo and Hf had a much slower inward diffusion rate and mainly enriched in the outer layer to form (Mo,Hf)Si_2_ phases, as shown in Figure 3a,d. At the same time, the solid Al_2_O_3_ particle would not participate in the chemical reaction and was distributed between the sintering reaction products on the coating surface, forming a relatively loose (Mo,Hf)Si_2_-Al_2_O_3_ outer layer on the surface, as shown in Figure 3a,e,f. The formation of pores and pits on the coating surface was related to the gaseous volatilization resulting from the decomposition of the nitrocellulose during the sintering process.

### 3.2. Oxidation Behavior of the Coating

Isothermal oxidation tests of the coating were conducted at 1400 °C in static air. The results show that the (Mo,Hf)Si_2_-Al_2_O_3_ composite coating exhibited excellent oxidation resistance and could provide effective oxidation protection for Mo-based alloy up to 40 h at 1400 °C. Figure 5 exhibits the weight gain of the coating as a function of duration time ranging from 0.25 to 40 h. It can be seen that the coating exhibited continuous weight gain with increasing oxidation time, containing a rapid weight gain stage (Stage I, ranging from 0 to 1 h) and a slow weight gain stage (Stage II, ranging from 1 to 40 h), as shown in Figure 5. At the early stage of oxidation (Stage I), the mass change within an approximately linear region 0–1 h was 2.34 mg/cm^2^, while the final weight gain rate of the coating after 40 h oxidation was only 6.03 mg/cm^2^ after 40 h of oxidation. Changes in weight obeying a parabolic law were observed for the (Mo,Hf)Si_2_-Al_2_O_3_ coating in this work and MoSi_2_-Al_2_O_3_ in our previous work [5].

The revealed regularities of oxidation of the coating were attributed to the oxidation behavior during oxidation. In the initial stage, the coating oxidized, and the oxide scale (containing, e.g., HfO_2_, SiO_2_) was formed on the coating surface, resulting in rapid weight gain of the coating. However, the oxide scale was insufficient to wrap the substrate; consequently, significant oxidative weight gain was observed in Stage I. In Stage II, a relatively dense oxide scale was formed on the coating surface with an oxidation time extension. Since a protective oxide scale formed on the surface of the coating, the oxidation rate was under the control of the permeation of oxygen through the dense oxide scale, thus the mass gain rate decreased. 

### 3.3. Phase Composition of the Oxidized Coating

Figure 6 shows the XRD patterns of the surface of the (Mo,Hf)Si_2_-Al_2_O_3_ coating after oxidation for different times, and the phase composition of the coating after oxidation is given in Table 2. Markers show positions of the peaks corresponding to MoSi_2_, Mo_5_Si_3_, mullite, Al_2_O_3_, HfO_2_, SiO_2_, and HfSiO_4_ after oxidation for 0.5 h. The phase composition of the coating surface after oxidation for 1 h and 5 h was the same as that of 0.5 h, while the intensity of diffraction peaks differed. At an exposure duration of 0.5–5 h, the diffraction peak intensity of MoSi_2_ in the oxidized coating gradually decreased, while the peak intensity of Mo_5_Si_3_, HfO_2_, SiO_2_, HfSiO_4_, and mullite increased with the extension of oxidation time, indicating that the content of MoSi_2_ phases gradually decreased, while the content of Mo_5_Si_3_, HfO_2_, SiO_2_, HfSiO_4_, and mullite increased. It is worth noting that the diffraction peak of MoSi_2_ was absent, and Mo_3_Si appeared significantly after 10 h of oxidation. The appearance of the Mo diffraction peak in the XRD pattern after 20 h of oxidation is supposed to be the Mo-alloy substrate. Compared with the XRD result of the unoxidized coating, it can be inferred that the presence of Mo_5_Si_3_, HfO_2_, and SiO_2_ was attributed to the oxidation of MoSi_2_ and (Mo,Hf)Si_2_. Meanwhile, the production of mullite and HfSiO_4_ is obviously resulting from the synthetic reactions between Al_2_O_3_, HfO_2_, and SiO_2_. The presence of Al_2_O_3_ and HfO_2_ phases during the whole 40 h oxidation demonstrates that only part of them reacted with SiO_2_ to form mullite and HfSiO_4_.

### 3.4. Microstructure Evolution of the Coating during Oxidation

#### 3.4.1. Surface Morphology

Figure 7a–f reveals the surface morphology of the composite coating after oxidation for the time ranging from 0.5 to 40 h. In comparison with the surface of the as-prepared coating, the composite coating after oxidation exhibited a relatively compact and smooth structure. Glassy phases were formed on the surface of the coating after oxidation for 0.5 h, and there were granular and flaky particles inlaying in the glassy phases. Meanwhile, a ceramic layer was observed near the glassy oxides, as shown in Figure 7a. Figure 8 shows the element mapping of the coating after 0.5 h of oxidation. It can be seen that the glassy phases were rich in Si and O (SiO_2_-based oxide), while the ceramic layer was rich in Al and O. The Hf element was mainly distributed in the glassy oxides. It can be inferred that the white particles (Spot 2 in Figure 1a, identified to be MoSi_2_ with Hf dissolved in it) were oxidized to form SiO_2_-based glassy oxides with HfO_2_ inclusions. The dark gray particles (Spot 1 in Figure 1a, identified to be Al_2_O_3_) were stable for oxygen. However, the small size of the white particles (silicides) was observed to be found in the cavity of Al_2_O_3_ particles (Figure 1a) and would oxidize to form SiO_2_-based oxides. Moreover, the Al_2_O_3_ layer on the coating surface was relatively loose, and oxygen could pass through the Al_2_O_3_ layer and react with silicide beneath the Al_2_O_3_ layer to generate SiO_2_-based oxides. The Al_2_O_3_ particles would further be likely to react with SiO_2_ to form mullite, leading to the formation of mountain-like clusters, which were marked by a purple outer oval, as shown in Figure 7a,b. Distinguished from the MoSi_2_-Al_2_O_3_ coating in our previous work [5], the SiO_2_-based glassy oxides formed on the surface of the (Mo,Hf)Si_2_-Al_2_O_3_ coating in this work were distributed uniformly and would not bond together to form a complete and dense glassy oxide film on the coating surface. Instead, ceramic phases (e.g., Al_2_O_3_, mullite) and glassy phases alternated on the coating surface, as shown in Figure 7a–c. The surface morphologies of the composite coating after oxidation for 1 h and 5 h are similar to that of 0.5 h. The amount and size of both granular and flaky particles increased with oxidation time. Figure 7d–f shows the surface morphology of the composite coating after exposure duration for a longer time (10, 20, and 40 h). Obviously, the coating surface had a tendency to ceramicize, and the SiO_2_-based glassy oxides were almost placed by a large number of island-like ceramic clusters, which illustrates that the glassy oxide was consumed. Surface morphology evolution (Figure 7) well corresponded with the XRD analysis results (Figure 6). It is believed that SiO_2_ reacted with pre-existed Al_2_O_3_ and newly generated HfO_2_ particles to form mullite and HfSiO_4_, respectively, according to the XRD result in Figure 6, which resulted in a rough surface of the oxide layer. It has been reported that the formation of mullite and HfSiO_4_ was beneficial to the anti-oxidation property resulting from their excellent thermal stability, low oxygen permeability, and the improvement of thermal mismatch between oxide and coating layers.

#### 3.4.2. Cross-Sectional Microstructure

Figure 9 shows the cross-sectional microstructure evolution of the composite coating after oxidation ranging from 0.5 to 40 h. It can be seen from these pictures that the composite coating exhibited a multi-phase structure. As shown in Figure 9a, a four-layer structure marked as Layer I, Layer II, Layer III, and Layer IV from inside to outside is observed in the cross-section of the coating after oxidation for 0.5 h, with a thickness of 14.3 μm, 30 μm, 35 μm, and 16.7 μm respectively. It is concluded that the thickness of the MoSi_2_ layer reduced by about 20 μm (from 50 μm to 30 μm), while the thickness of the Mo_5_Si_3_ layer increased by about 14.3 μm (from 0 to 14.3 μm), compared to the as-prepared coating. Figure 10 shows the element mapping of the cross-section of the oxidized coating, where it is indicated that Layer I was an oxide scale mainly containing Si, Al, Hf, and O elements, while Layer II was a silicide–oxide composite layer mainly containing Si, Mo, Al, Hf, and O elements. It seems that Layers III and IV both mainly contained Si and Mo elements, of which Layer III was rich in Si while Layer IV was Si-poor Mo-silicide. Table 3 gives the average chemical composition of Spots 1–9 in Figure 9. According to the atomic ratios of the elements and the XRD pattern (Figure 6), Spots 1–5 were Mo_5_Si_3_, MoSi_2_, (Mo,Hf)Si_2_, (Mo,Hf)_5_Si_3_, and Si-Al-Hf-O oxide (a mixture of SiO_2_, Al_2_O_3_, and HfO_2_), respectively. Consequently, Layers I, II, III, and IV were identified to be Mo_5_Si_3_, MoSi_2_, MoSi_2_+Mo_5_Si_3_, and Si-Al-Hf-O oxide, respectively. A thick Mo_5_Si_3_ diffusion layer was formed between the MoSi_2_ layer (Layer II) and the substrate, and a small amount of Mo_5_Si_3_ phase existed between the oxide scale and the MoSi_2_ layer. The thin Mo_5_Si_3_ layer formed between the oxide scale and MoSi_2_ layer was caused by the oxidation of MoSi_2_ (Si diffusion outward to react with oxygen), while the Mo_5_Si_3_ layer (Layer I in Figure 9a) between the substrate and MoSi_2_ layer was formed due to decomposition of MoSi_2_ (Si diffusion from the coating to the substrate). The coating structures after oxidation at 1400 °C for 1 h (Figure 9b) and 5 h (Figure 9c) also consisted of four layers, which is similar to that of 0.5 h. Moreover, the phase composition of each layer was also the same, while the thickness differed. The thickness of the MoSi_2_ layer (Layer II) reduced to 31.3 μm, while the Mo_5_Si_3_ layer (Layer III) and the oxide scale (Layer IV) thickened to 25 μm and 16.6 μm, respectively, after 1 h of oxidation. The value of the thickness of the MoSi_2_ layer, Mo_5_Si_3_ layer, and oxide layer was 24.3 μm, 51.5 μm, and 21.4 μm, respectively, when the oxidation time reached 5 h. The results illustrate the continuous consumption and decomposition of MoSi_2_ during oxidation, giving rise to the increase in the thickness of the Mo_5_Si_3_ and SiO_2_-based scale. It can also be seen from Figure 9a–f that a large amount of HfO_2_, HfSiO_4_, Al_2_O_3_, and mullite was distributed in the outer layer of the oxide scale (Layer IV), while a small part of them was distributed in the silicide layer (Layer III), as shown in Figure 9a–c. Concurrently, the amount of white particles in the outer layer of the oxide scale (Layer IV) continued to increase with the extension of oxidation time, especially after 20 and 40 h of oxidation, as shown in Figure 9e,f. This is attributed to the persistent oxidation of Hf to form HfO_2_, and HfO_2_ would further generate HfSiO_4_ by reacting with SiO_2_, which could be confirmed by the XRD analysis. HfO_2_, HfSiO_4_, Al_2_O_3,_ and mullite particles inlaid into the SiO_2_-based oxide scale and could form a skeleton structure, in which HfO_2_, HfSiO_4_, Al_2_O_3,_ and mullite particles play a pinning role on the SiO_2_-based oxide scale and could enhance the structural stability of the oxide scale. 

Figure 11 shows an enlarged view of the interface between the oxide scale and coating after 5 h of oxidation. A ceramic layer with a large amount of oxides (e.g., Al_2_O_3_, HfO_2_) and ceramic particles (e.g., mullite, HfSiO_4_) was observed outside the oxide scale. Mo_5_Si_3_ and MoSi_2_ were closely bonded. SiO_2_-based oxides containing Al_2_O_3_ and HfO_2_ were embedded in the gap of the Mo_5_Si_3_ and MoSi_2_ phases. Simultaneously, HfO_2_ particles were partly dispersed in the MoSi_2_ phases, which illustrates the internal oxidation of (Mo,Hf)Si_2_. The growth and agglomeration of HfO_2_ particles can also be observed. The HfO_2_ particles had a maximum size of ~5.5 μm. The distribution of Al_2_O_3_ particles is similar to that of HfO_2_. 

It is worth noting that, as the oxidation time was further extended to 10 h, although the coating still displayed a four-layer structure, significant changes had taken place in the thickness and composition of each layer of the coating. To distinguish the structure of the coating before and after oxidation for 10 h, the four layers in Figure 9d are marked as Layer i, Layer ii II, Layer iii, and Layer iv from inside to outside. According to the EDS (Table 3) and XRD results (Figure 6), it can be inferred that the phase composition of Layers i, ii, and iv was Mo_3_Si, Mo_5_Si_3_, and Si-Al-Hf-O oxide, with a thickness of 10.7 μm, 81.1 μm, and 24.9 μm, respectively, while Layer iii was a composite layer mixture of Mo_5_Si_3_ and Si-Al-Hf-O oxides. The MoSi_2_ layer was absent after 10 h of oxidation, which is consistent with the XRD analysis in Figure 6. These above results indicate the complete consumption of the MoSi_2_ layer. As compared to the microstructure of the coating after oxidation for 5 h, a Mo_3_Si layer was newly generated between the Mo_5_Si_3_ layer (Layer ii) and the substrate after 10 h of oxidation. However, no Mo_3_Si formation was observed between the Mo_5_Si_3_ layer (Layer ii) and the oxide scale (Layer iv). The coating structures after 20 h and 40 h of oxidation were similar to that of 10 h oxidation. The thickness of the Mo_3_Si layer (Layer i) and oxide scale both increased with oxidation time. 

Figure 12 displays the thickness variation of coating layers after oxidation. It is obvious that the oxide layer was growing constantly during the exposure duration of 0.5–40 h. During the whole oxidation process, the thickness of the Mo_5_Si_3_ layer increased gradually and reached the maximum (81.1 μm) after 10 h of oxidation, then began to decrease with the increase in oxidation time. When the oxidation time reached 40 h, the thickness of the Mo_5_Si_3_ layer decreased to 75.7 μm. The thickness of the Mo_3_Si layer (Layer i) increased to 17.1 μm and 21.4 μm, and the thickness of the oxide scale increased to 25.5 μm and 29.7 μm, respectively, after 20 h and 40 h of oxidation.

According to the element mapping of the coating after oxidation in Figure 10, it is concluded that during oxidation, Hf and Al elements tended to concentrate on the silicide–oxide composite layer (Layer III and iii) and the oxide scale (Layer IV and iv), which reveals that HfO_2_, HfSiO_4_, Al_2_O_3_, and mullite particles were mainly distributed in the oxide scale and silide–oxide composite layer. This is consistent with the cross-sectional morphology observation in Figure 9. 

Microcracks were found in the MoSi_2_ layer after 0.5 h of oxidation and reached the Mo_5_Si_3_/MoSi_2_ interface after 1 h of oxidation. When the oxidized samples were cooled down from the test temperature to room temperature, cracks could be produced induced by large thermally induced stresses. After long-time oxidation (20 h and 40 h), cracks passed through the Mo_3_Si layer and propagated to the Mo_3_Si/substrate interface. Simultaneously, there were some Kirkendall voids in the Mo_5_Si_3_ layer after oxidation for 1 h, which were marked with a blue circle, and the number of Kirkendall voids increased with the extension of oxidation time to form porous zones in the Mo_5_Si_3_ layer, as shown in Figure 9b–f. The occurrence of voids in the Mo_5_Si_3_ layer was attributed to the faster diffusion of Si than that of Mo in the Mo_5_Si_3_ phase [5]. The mismatch of the CTE between the oxide scale and the MoSi_2_ layer easily leads to the cracking and shedding of the oxide scale. However, no visible cracks were revealed in the oxide scale after 10 h of oxidation in this work, indicating that the oxide scale containing Al_2_O_3_, HfO_2_, mullite, HfSiO_4_. and SiO_2_ exhibited enhanced thermal matching with the MoSi_2_ layer. Nonetheless, cracks were formed in the Mo_3_Si layer after 20 h of oxidation, as shown in Figure 9e,f. It is confirmed that the Mo_5_Si_3_ layer (Layer I in Figure 9a–c) and the Mo_3_Si layer (Layer i in Figure 9d–f) were beneficial to prevent cracks propagating toward the Mo-based alloy substrate since no cracks were found to propagate into the matrix even after 40 h of oxidation [5,39]. In addition, It can also be seen that even after oxidation at 1400 °C for 40 h, the (Mo,Hf)Si_2_-Al_2_O_3_ composite coating remained intact, which indicates that the composite coating was still efficiently protective.

### 3.5. Antioxidation Mechanism of the Composite Coating

Figure 13 displays the schematic oxidation mechanism of the (Mo,Hf)Si_2_-Al_2_O_3_ composite coating at 1400 °C in air. The composite coating exhibits a double-layer structure consisting of a MoSi_2_ inner layer and an outer layer of a mixture of (Mo,Hf)Si_2_ and Al_2_O_3_ (Figure 13a). Reactions (2)–(9) may occur when the coating samples are exposed to a high-temperature oxidizing environment at 1400 °C. The standard Gibbs free energy (based on 1 mol oxygen) of Reaction (2) is lower than that of Reaction (3), which is −420.27 kJ/mol and −559.84 kJ/mol, respectively [39,40]. Similarly, the standard Gibbs free energy of Reaction (4) is lower than that of Reaction (5). Thus, Reactions (2) and (4) are dominant. In the initial stage of oxidation, the oxidation of MoSi_2_ and (Mo,Hf)Si_2_ gives rise to form SiO_2_, HfO_2_, Mo_5_Si_3_, and MoO_3_ on the coating surface according to Reactions (2)–(5). A thin Mo_5_Si_3_ layer could be observed between the oxide scale and the coating (Figure 9a and Figure 13b). In addition to Reactions (2)–(5), the pre-existing Al_2_O_3_ and newly generated HfO_2_ were partly converted to mullite and HfSiO_4_, respectively, according to Reactions (6) and (7). Since MoO_3_ would volatilize rapidly at the temperature for the isothermal oxidation test in this work, a glassy SiO_2_-based oxide scale with HfO_2_, Al_2_O_3_, mullite, and HfSiO_4_ inclusions was formed on the coating surface. Since the silicide–oxide composite layer (Layer II in Figure 3a) was relatively loose, oxygen could also diffuse inward through the pores and pits directly and react with MoSi_2_ and (Mo,Hf)Si_2_ in the sublayer (Layer II in Figure 3a). Thus, glassy SiO_2_-based oxides were also formed inside the silicide–oxide composite layer. Al_2_O_3_ particles embedded in the MoSi_2_ phases in the as-prepared coating (Figure 3a) were then embedded in SiO_2_-based oxides (Figure 10) along with the MoSi_2_ transformation to SiO_2_ during oxidation. With the formation of the oxide scale, the pores and pits could be filled, and the inward diffusion of oxygen to the coating was hindered.

Compared with the MoSi_2_-Al_2_O_3_ composite coating in our previous work [5], the composite coating in this work did not generate a complete and dense glassy SiO_2_-based oxide scale to cover the whole coating surface after 0.5 h of oxidation. Instead, a large amount of white particles (e.g., HfO_2_, HfSiO_4_, Al_2_O_3_, mullite) were concentrated on the coating surface to form a ceramic outer layer. This phenomenon was partly related to the preferential oxidation of Hf. Hf shows a higher affinity with oxygen relative to Si and could be oxidized preferentially at high temperatures. Through that, the reaction of Si with oxygen to form SiO_2_ could be relatively suppressed, and the driving force for its outward diffusion could be decreased as well. Hence, a large amount of white particles (e.g., HfO_2_) were agglomerated and enriched in the outer layer of the oxide scale, which is confirmed by the surface morphology of the oxidized coating (Figure 6). Since Al_2_O_3_ particles were distributed mainly in the outer layer in the as-prepared coating, they would distribute mainly in the outer layer of the oxide scale in the oxidized coating. Furthermore, Reactions (6) and (7) concerning the formation of mullite and HfSiO_4_ consumed a certain amount of SiO_2_, which led to the coating failing to form a glassy oxide film properly at the place where HfO_2_ and Al_2_O_3_ particles were enriched. However, a dense oxide film was formed below, acting as a diffusion barrier of oxygen, as shown in Figure 9a–f. Meanwhile, the formation of the composite oxide scale embedded with high-temperature stable mullite and HfSiO_4_ is believed to improve the high-temperature resistance of the composite coating because of the following factors: (1) Al_2_O_3_, mullite, HfO_2_, HfSiO_4_, and SiO_2_ could develop a special glass–ceramic skeleton structure on the coating surface and possessed the peculiarities of the stress tolerance of glass scale and the structural stability of the ceramic phase, in which Al_2_O_3_, HfO_2_, mullite, and HfSiO_4_ particles have a pinning effect on the SiO_2_ glassy oxide [20]. (2) The permeability of oxygen through this composite oxide scale is supposed to be lower than that of oxygen in the SiO_2_ scale [41]. (3) The CTE of Al_2_O_3_ (8.3 × 10^−6^/K), HfO_2_ (5.8 × 10^−6^/K), mullite (5.6 × 10^−6^/K), and HfSiO_4_ (4.6 × 10^−6^/K) is higher than that of SiO_2_ (0.55 × 10^−6^/K). The CTE of the composite oxide scale is higher than that of the pure SiO_2_ scale and closer to that of the coating, which is expected to improve the CTE mismatch between the oxide scale and the coating. For the above reasons, the composite oxide scale could be well stabilized on the coating surface and provide long-term oxidation protection for Mo-based substrates by retarding oxygen diffusion into the internal coating and suppressing crack propagation in the oxide scale.

In addition to the surface layer, the microstructure inside the coating also changed. A diffusion layer of Mo_5_Si_3_ was formed between the MoSi_2_ layer and substrate according to Reaction (8), as shown in Figure 13b. It is believed that the formation of the thin Mo_5_Si_3_ layer between the oxide scale and MoSi_2_ layer was attributed to the oxidation of MoSi_2_, while the Mo_5_Si_3_ layer between the MoSi_2_ layer and the substrate was derived from the inward diffusion of Si (MoSi_2_) from coating to substrate according to Reaction (8) rather than oxidation [5]. During oxidation, Si atoms diffuse outward to generate Mo_5_Si_3_ and SiO_2_, and inward to the substrate to form Mo_5_Si_3_ (Layer I); at the same time, Hf would diffuse outward and react with oxygen to form HfO_2_.

With the extension of oxidation time, the continuous consumption of MoSi_2_ led to a gradual decrease in thickness, accompanied by an increase in the thickness of the Mo_5_Si_3_ layer and the oxide scale layer. When the oxidation time reached 10 h, the MoSi_2_ layer was absent (Figure 13d) and a diffusion layer of Mo_3_Si was generated between the tMo_5_Si_3_ layer and the substrate. It is believed that the formation of Mo_3_Si was attributed to the diffusion of Si from Mo_5_Si_3_ to the Mo-based substrate driven by the concentration gradient according to Reaction (9).

The outward diffusion of Si and Hf from the coating to the oxide scale and the interdiffusion between the coating and substrate during oxidation resulted in a multi-layer structure of the coating after oxidation. Since the thermal stress for the Mo_5_Si_3_/Mo interface is lower than that for the MoSi_2_/Mo substrate [5], the formation of the multi-layer structure of the coating after oxidation is beneficial to release the residual stress induced by thermal mismatch between the composite coating and the substrate [42,43], which would restrain the crack formation in the coating and prevent crack propagation toward the substrate even after 40 h of oxidation (Figure 9f and Figure 13d).
5/7MoSi_2_ + O_2_ = 1/7Mo_5_Si_3_ + SiO_2_(2)
2/7MoSi_2_ + O_2_ = 2/7MoO_3_ + 4/7SiO_2_(3)
(Mo,Hf)Si_2_ + O_2_ →Mo_5_Si_3_ + HfO_2_ + SiO_2_(4)
(Mo,Hf)Si_2_ + O_2_ → MoO_3_ + HfO_2_ + SiO_2_(5)
3Al_2_O_3_ + 2SiO_2_ = 3Al_2_O_3_·2SiO_2_ (mullite)(6)
HfO_2_ + SiO_2_ = HfSiO_4_(7)
MoSi_2_ + 7/3Mo = 2/3Mo_5_Si_3_(8)
Mo_5_Si_3_ + 4Mo = 3Mo_3_Si(9)

## 4. Conclusions

In this study, a highly oxidation-resistant MoSi_2_-based composite coating with Hf and Al_2_O_3_ co-doping was prepared by the slurry sintering method. Additionally, its oxidation behavior was systematically investigated at 1400 °C. The coating was composed of MoSi_2_ with Hf existing in solid solution and Al_2_O_3_ phases and had a double-layer structure consisting of a MoSi_2_ inner layer and a (Mo,Hf)Si_2_-Al_2_O_3_ outer composite layer. The composite coating exhibited excellent oxidation resistance at 1400 °C, and the mass gain was only 6.03 mg/cm^2^ after 40 h of oxidation. During oxidation, Hf diffused outward and underwent preferential oxidation to generate HfO_2_, while Si was susceptible to diffuse outward to form SiO_2_, Mo_5_Si_3_ on the coating surface by oxidation of MoSi_2_ and inward to form Mo_5_Si_3_ diffusion layer between coating and substrate by decomposition of MoSi_2_. Pre-existing Al_2_O_3_ and newly generated HfO_2_ particles partly reacted with SiO_2_ to form thermal stable mullite and HfSiO_4_. SiO_2_, Al_2_O_3_, HfO_2_, mullite, and HfSiO_4_ developed a dense protective oxide scale on the surface of the composite coating during oxidation. The composite oxide scale developed a skeleton structure with enhanced thermal stability and improved adhesion between the oxide scale and coating, which enhanced the thermal stability of the oxide scale. Simultaneously, element (e.g., Si, Hf) diffusion led to the formation of a multi-layer structure in the coating, which is beneficial to release the residual stress induced by thermal mismatch between the coating and the substrate and inhibit the tendency of crack formation or prevent crack propagation directly on the substrate.

## Figures and Tables

**Figure 4 materials-16-03215-f004:**
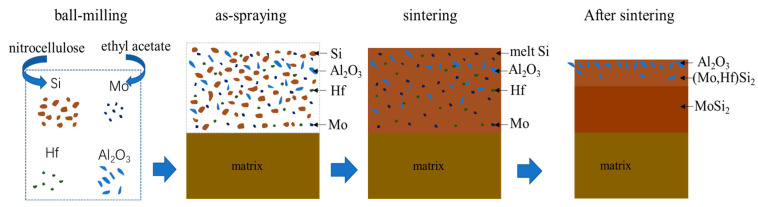
Schematic diagram of microstructure evolution during coating preparation.

**Figure 5 materials-16-03215-f005:**
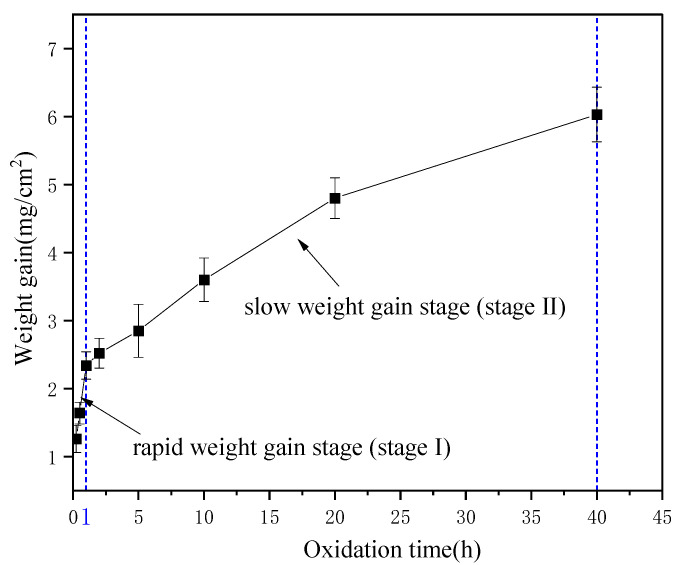
Weight gain rate of the coating as a function of duration times.

**Figure 6 materials-16-03215-f006:**
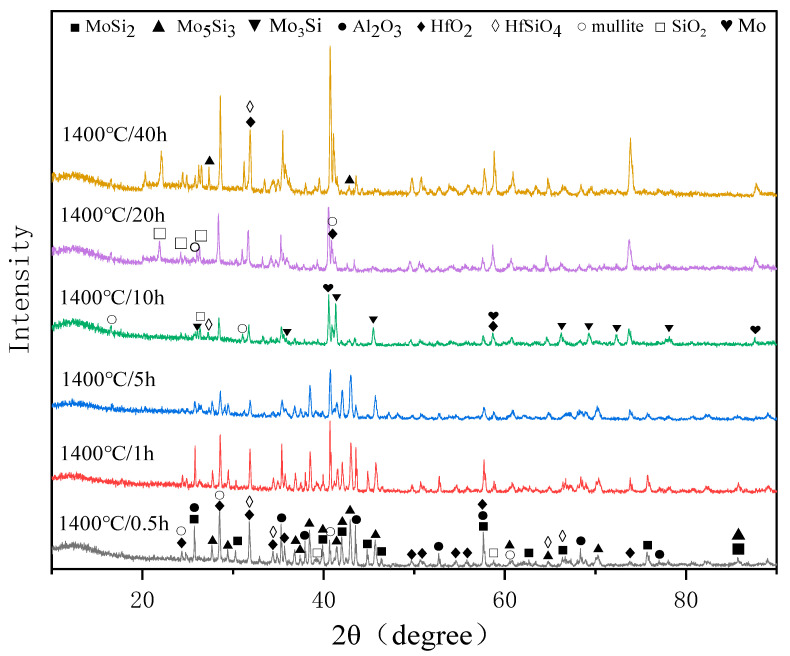
XRD patterns of the surface of the oxidized coatings.

**Figure 7 materials-16-03215-f007:**
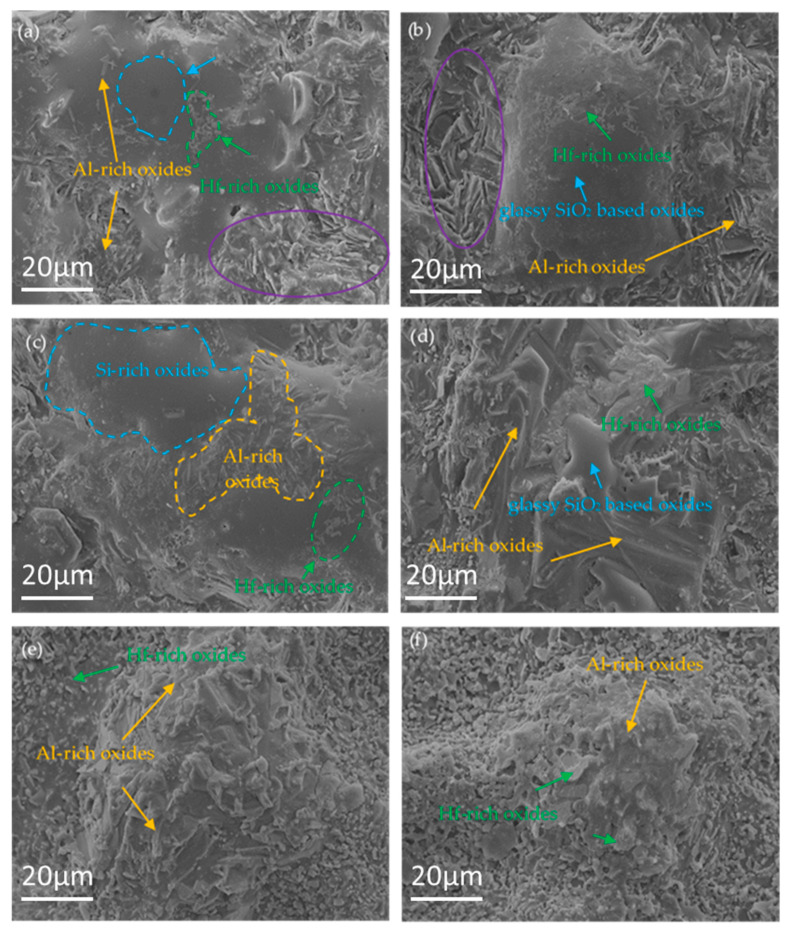
Surface morphology of the coating after oxidation for different times: (**a**) 0.5 h; (**b**) 1 h; (**c**) 5 h; (**d**) 10 h; (**e**) 20 h; (**f**) 40 h.

**Figure 8 materials-16-03215-f008:**
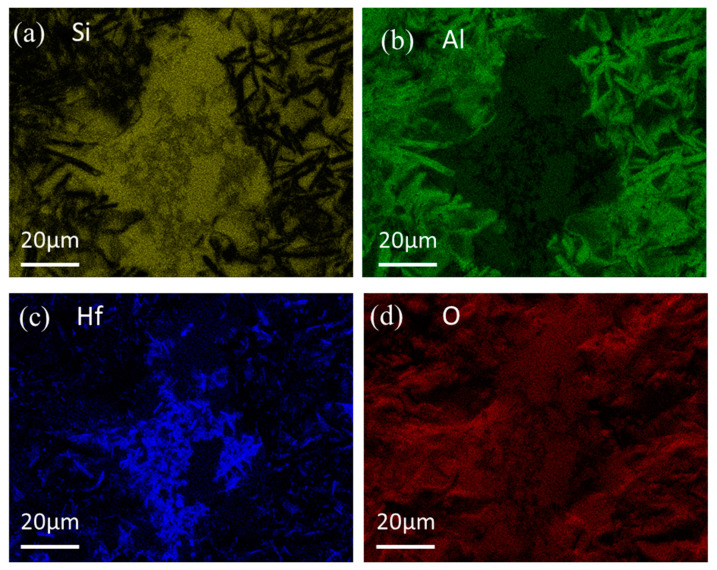
Element mapping of the surface of the coating after 0.5 h of oxidation (**a**) Si; (**b**) Al; (**c**) Hf; (**d**) O.

**Figure 9 materials-16-03215-f009:**
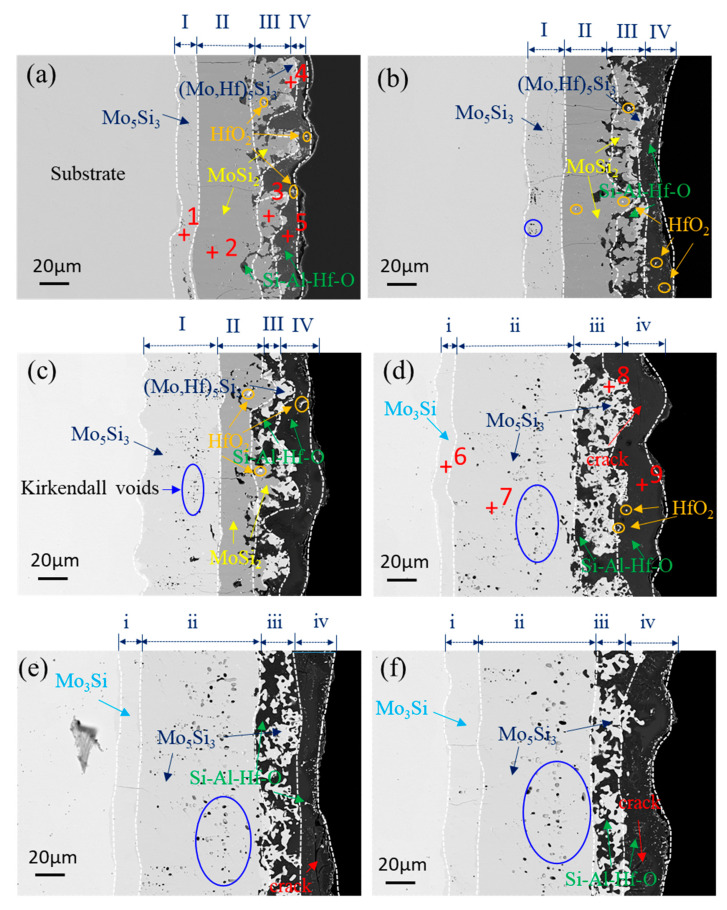
Cross-sectional microstructure of the coating after oxidation for different times: (**a**) 0.5 h; (**b**) 1 h; (**c**) 5 h; (**d**) 10 h; (**e**) 20 h; (**f**) 40 h.

**Figure 10 materials-16-03215-f010:**
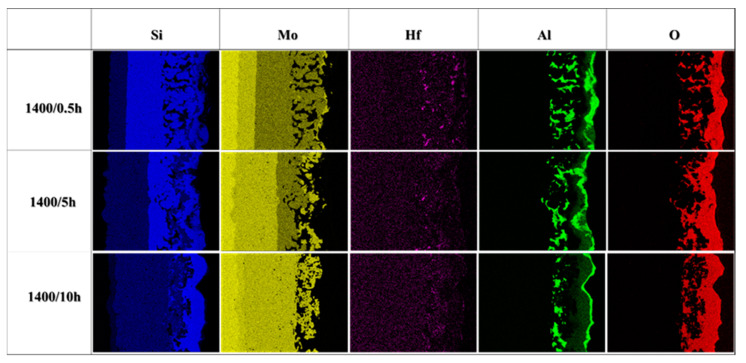
Element mapping of the cross-sectional of the oxidized coating.

**Figure 11 materials-16-03215-f011:**
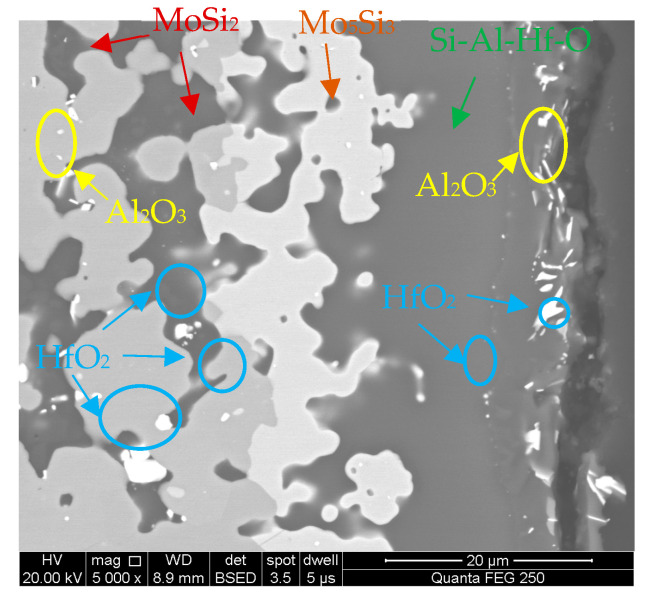
An enlarged view of the coating/oxide scale interface after 5 h oxidation.

**Figure 12 materials-16-03215-f012:**
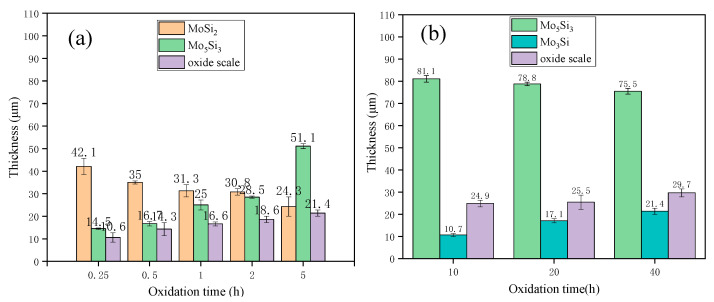
Thickness of coating layers after oxidation for (**a**) 0–5 h and (**b**) 10–40 h.

**Figure 13 materials-16-03215-f013:**
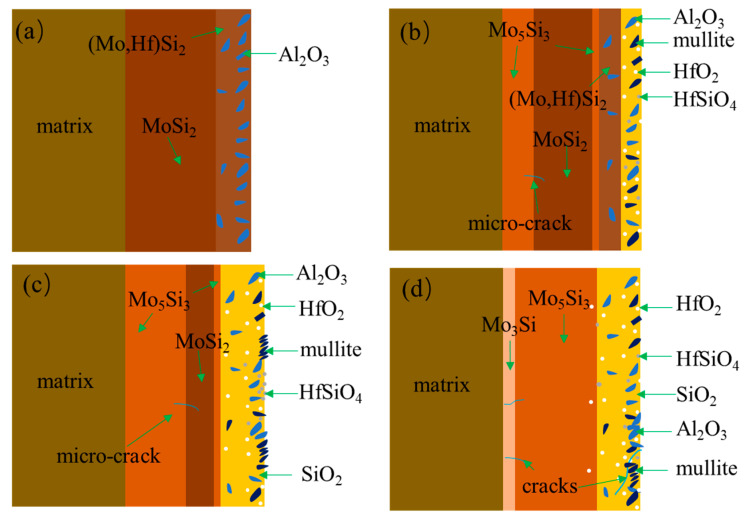
Schematic oxidation mechanism of the composite coating at 1400 °C in air (**a**) as-prpared coating, (**b**) 1 h of oxidation; (**c**) 5 h of oxidation; (**d**) 10 h of oxidation.

**Table 2 materials-16-03215-t002:** Phase composition of the composite coating after oxidation at different times.

Time (h)	Phase Composition
0.5, 1, 5	MoSi_2_, Mo_5_Si_3_, mullite, Al_2_O_3_, HfO_2_, SiO_2_, HfSiO_4_
10, 20, 40	Mo, Mo_5_Si_3_, Mo_3_Si, HfO_2_, Al_2_O_3_, SiO_2_, mullite, HfSiO_4_

**Table 3 materials-16-03215-t003:** EDS analysis of Spots 1–9 in Figure 9.

Spot	Composition (at.%)					Main Phase
	Mo	Hf	Si	Al	O	
1	58.92	-	39.84	-	1.24	Mo_5_Si_3_
2	31.68	0.12	66.91	-	1.29	MoSi_2_
3	29.69	0.26	66.93	-	3.12	MoSi_2_ (Hf)
4	55.87	-	40.02	-	4.11	Mo_5_Si_3_
5	-	0.95	12.14	32.98	53.93	SiO_2_, Al_2_O_3_, HfO_2_
6	62.20		23.88	-	13.92	Mo_3_Si
7	53.65	-	36.46	-	09.89	Mo_5_Si_3_
8	65.81	-	25.54	-	08.65	Mo_3_Si
9	-	0.54	35.73	6.97	56.76	SiO_2_, Al_2_O_3_, HfO_2_

## Data Availability

The data presented in this study are available on reasonable request from the corresponding author.

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
