# Peer review of "Oxidation Behavior of (Mo,Hf)Si2-Al2O3 Coating on Mo-Based Alloy at Elevated Temperature"

_materials, 2023, doi:10.3390/ma16083215_

Round 1

Reviewer 1 Report

The manuscript entitled “Oxidation behavior of (Mo,Hf)Si2-Al2O3 coating on Mo-based alloy at elevated temperature” is well written. The methods are reproducible. 

In this manuscript, Hf and Al2O3 were co-doped to MoSi2 coating to form a new (Mo,Hf)Si2- 69 Al2O3 composite coating on Mo-based alloy.  Hf-Al2O3 co-doped MoSi2 coating was fabricated on the Mo-based substrate by a method of slurry sintering. The composite coating was evaluated at 1400 ℃. The authors also discussed the microstructure evolution, the element distribution, and the phase composition of the coatings before and after oxidation, as well as the antioxidant mechanism of the coating at high temperature.

In my opinion, a few comments needed to be amended to improve the quality of this manuscript. The comments are below:

Title: Oxidation behavior of (Mo,Hf) Si2-Al2O3 coating on Molybdenum-based alloy at elevated temperature

1.               Please send this manuscript to the proofreading service. Some errors and typos are detected.

2.               Please write full name of the alloy before its abbreviation e.g. Molybdenum (Mo), Hf?

3.               Figure 1, please enlarge the scale bar, Figure 2. (b-f) , Figure 7, Figure 8- please include the scale bar.

4.               Figure 2. (a)XRD pattern- please enlarge this figure.

5.               Please enlarge the size of Figure 4 and Figure 6

6.               Figure 5, please include error bars.

7.               Please improve the figure legend of all figures so they can be self-explanatory.

Kindly send this manuscript to any proofreading service to improve the quality. 

Author Response

Dear Reviewer

Thank you very much for your time involved in reviewing the manuscript and your very encouraging comments on the manuscript.

We also appreciate your clear and detailed feedback and hope that the explanation has fully addressed all of your concerns. In the remainder of this letter, we discuss each of your comments individually along with our corresponding responses. To facilitate this discussion, we first retype your comments in italic font and then present our responses to the comments.

We would like to take this opportunity to thank you for all your time involved and this great opportunity for us to improve the manuscript. We hope you will find this revised version satisfactory.

sincerely

Li Wei

Reviewer 2 Report

Review on "Oxidation behavior of (Mo,Hf)Si2-Al2O3 coating on Mo-based 

alloy at elevated temperature " by Lv et al.

This paper reports the formation of a new coating based on the doped MoSi2 and Al2O3 to cover the metal surface and prevent its high-temperature oxidation.
The introduction section provides a clear rationale behind this work. It was described and explained, why this alloying of silicide coating by Hf and Al2O3 may be advantageous. 
The authors performed a versatile and well-developed study. The coating was synthetised, examined by XRD, SEM, EDS and EPMA techniques. The oxidation behaviour of the coating was melticulously studies within different time intervals. The comparison of the results with those in absence of Hf-alloying was performed.
In general, this is a very good paper. The reviewer has only a couple of minor questions.
1) Why the Mo-La2O3 alloy was used as the substrate? Does the proposed coating able to protect other alloy types (including non-molybdenum alloys) from oxidation?

2) Why the ration of 65:20:5:10 for Si: Mo: Hf: Al2O3 was used?
Did the authors investigate, how much Hf could dissolve in MoSi2? Is there any "sufficient" and "optimal" amount of doping Hf to form a stable solid solution? Could the authors discuss the influence of the amount of Hf on the properties of the obtained coating? E. g., from the Table 2 it is evident that the Hf/Mo ration in the silicide phase is about 1:15. What happens, if another amount of Hf would have been added to the coating, how the properties could be affected?

3) Figure 5. It seems, even after 40h of oxidation the process is not inhibited completely and still continues. The density of the layer is not enough to prevent inward diffusion of oxygen or outward diffusion of the metal to the surface. It is very interesting to reach the equilibrium. From the present figure it could be assumed that the oxidation will slowly progress until the substrate itself will be fully oxidised, and the coating protection ability is not so effective to block the oxidation completely.

4) Table 3. What is the "Si-Hf-Al-O oxide"? Is it a mixture of simple oxides, or a complex silicate?

5) Section 3.5. Please, provide the source (the reference data) for the Gibbs energies of reactions (2)-(9).

6) Reactions (4) and (5) are rather unambiguous. Without a clear content of Hf in the solid solution it is not possible to equalise them.

Best regards,

The reviewer.

None.

Author Response

Dear Reviewer

Thank you very much for your time involved in reviewing the manuscript and your very encouraging comments on the manuscript.

We also appreciate your clear and detailed feedback and hope that the explanation has fully addressed all of your concerns. In the remainder of this letter, we discuss each of your comments individually along with our corresponding responses. To facilitate this discussion, we first retype your comments in italic font and then present our responses to the comments.

We would like to take this opportunity to thank you for all your time involved and this great opportunity for us to improve the manuscript. We hope you will find this revised version satisfactory.

Comment 1: Why the Mo-La2O3 alloy was used as the substrate? Does the proposed coating able to protect other alloy types (including non-molybdenum alloys) from oxidation?

Response 1:Due to its high melting point, good conductivity and superior corrosion resistance, molybdenum (Mo) has been expected to be used in boats, heating elements, insulation screens and other fields. However, pure Mo suffers from low temperature brittleness, recrystallization brittleness and poor high temperature oxidation resistance which limits its application.

Adding rare earth oxides to Mo is an effective way to solve the above problems. Among those rare earth oxides, La2O3 has been considered the best additive. Mo-La2O3 alloy performs higher recrystallization temperature (>1400 ℃), improved strength and toughness and better oxidation resistance compared to pure Mo. However, poor high temperature oxidation resistance has still been a key bottleneck for wider application of Mo-La2O3 alloy at high temperatures. Therefore, we developed a novel Si-Mo-Hf-Al2O3 oxidation resistant coating on Mo-La2O3 alloy by a method of slurry sintering. The coating was formed through high temperature diffusion reactions between coating raw materials and substrate and had a metallurgical bonding with the substrate. This coating can not only be applied to Mo-based alloy, but also to other refractory metals and alloys (e.g., Nb-, Ta-, W- based alloy).

Comment 2: Why the ration of 65:20:5:10 for Si: Mo: Hf: Al2O3 was used?

Did the authors investigate, how much Hf could dissolve in MoSi2? Is there any "sufficient" and "optimal" amount of doping Hf to form a stable solid solution? Could the authors discuss the influence of the amount of Hf on the properties of the obtained coating? E. g., from the Table 2 it is evident that the Hf/Mo ration in the silicide phase is about 1:15. What happens, if another amount of Hf would have been added to the coating, how the properties could be affected?

Response 2: The content of Hf has been optimized and the ration of 65:20:5:10 for Si: Mo: Hf: Al2O3 was optimal in our experiment. We investigated the effect of Hf (0, 5, 10, 15, 20) on the oxidation resistance of the Si-Mo-Hf-Al2O3 composite coating, and found that Si-Mo-Al2O3 coating doped with Hf performed better oxidation resistance than that without Hf addition, and the antioxidation performance of the coating improved with the increase of Hf content (0~10%).

However, when the content of Hf is excessive (>10%), a large amount of HfO2 would be produced, and a great part of those HfO2 particles would react with SiO2 to form HfSiO4, causing a sharp decrease in the content of SiO2 in the oxide scale. HfO2 and HfSiO4 still agglomerate as solid particles in oxide scale at 1400 °C. The microscopic and porous HfO2 will become the fast channel for oxygen to enter the coating and accelerate the coating consumption. In addition, due to the low content of self-healing SiO2, oxide scale cannot promptly repair the coating defects and O quickly penetrates into the coating through defects such as cracks. Hence, excessive Hf will significantly reduce the coating life.

Comment 3: Figure 5. It seems, even after 40h of oxidation the process is not inhibited completely and still continues. The density of the layer is not enough to prevent inward diffusion of oxygen or outward diffusion of the metal to the surface. It is very interesting to reach the equilibrium. From the present figure it could be assumed that the oxidation will slowly progress until the substrate itself will be fully oxidised, and the coating protection ability is not so effective to block the oxidation completely.

Response 3: Bare Mo-based alloy suffers poor oxidation at high temperature oxidizing oxidizing (known as pest oxidation). The application of MoSi2-based coating could provide oxidation resistant protection for Mo-based alloy at high temperatures, attributing to the formation of a SiO2-based oxide scale on the coating surface by oxidation of MoSi2. The diffusion rate of oxygen through the SiO2-based oxide scale is extremely low, thus the diffusion of oxygen through the coating slow down sharply. The application of oxidation resistant coating is to restain the diffusion of oxygen to the substrate; however, it could not completely prevent oxidation. In our experiment, the Si-Mo-Hf-Al2O3 composite coating could could protect the substrate effectively at 1400 â—¦C for 120 h without failure, and enables the alloy to be applied in high temperature oxidizing environments.

Thank you for your inspiring comment. We are conducting a new research plan to apply a diffusion barrier layer between coating and substrate, which is considered to be effective to further retard the inward diffusion of oxygen or outward diffusion of the metal.

Comment 4: Table 3. What is the "Si-Hf-Al-O oxide"? Is it a mixture of simple oxides, or a complex silicate?

Response 4: We apologize for the inaccurate description. According to XRD and EDS results, it is a mixture of oxides consisting of SiO2, Al2O3 and HfO2. This inaccurate expression has been revised in the paper in Table 3.

Comment 5: Section 3.5. Please, provide the source (the reference data) for the Gibbs energies of reactions (2)-(9)

Response 5: Reactions (2) and (3) were possible reactions during oxidation of MoSi2. The Gibbs energies of reactions (2) and (3) were calculated by our team in previous work (Ref.40, 41), and the reference data has been cited in this work (in line 389).

Comment 6: Reactions (4) and (5) are rather unambiguous. Without a clear content of Hf in the solid solution it is not possible to equalise them

Response 6:Reactions (4) and (5) has been revised, as shown in lines 465 and 466 in the paper.

Reviewer 3 Report

The paper “Oxidation behavior of (Mo, Hf)Si2-Al2O3 coating on Mo-based alloy at elevated temperature, materials--2348775”, considered for publication in the Materials, presents the study of a highly oxidation-resistant MoSi2-based composite with Hf and Al2O3 co-doping. The work shows an excellent and complete structure. I suggest some corrections, which should be taken into consideration before publication.

1 - Question: The synthesis process is based on some other work? If so, cite.

2 - Question: Cite in the text the previous works mentioned in lines 249 and 411.

3 - Question: Present in the text the calculations for the Gibbs energy for equations 2 to 9, lines 467 to 474, as the values are being cited in the text.

Minor editing of English language required.

Author Response

Dear Reviewer

Thank you very much for your time involved in reviewing the manuscript and your very encouraging comments on the manuscript. We would like to take this opportunity to thank you for all your time involved and this great opportunity for us to improve the manuscript. We hope you will find this revised version satisfactory.

We also appreciate your clear and detailed feedback and hope that the explanation has fully addressed all of your concerns. In the remainder of this letter, we discuss each of your comments individually along with our corresponding responses.

To facilitate this discussion, we first retype your comments in italic font and then present our responses to the comments.

Comment 1: The synthesis process is based on some other work? If so, cite.

Response 1: The synthesis proces of Si-Mo-Hf-Al2O3 composite coating in this work was the same as that of Si-Mo-Al2O3 composite coating in our previous work (https://doi.org/10.1016/j.ceramint.2021.12.310). The previous works has been cited in the manuscript in lines 94 and 95.

Comment 2: Cite in the text the previous works mentioned in lines 249 and 405.

Response 2: Modifications have been made according to this requirement. The previous works was cited in lines 245 and 411 in the revised version.

Comment 3: Present in the text the calculations for the Gibbs energy for equations 2 to 9, lines 467 to 474, as the values are being cited in the text.

Response 3: Reactions (2) and (3) were possible reactions during oxidation of MoSi2. The Gibbs energies of reactions (2) and (3) were calculated by our team in previous work (Ref.40, 41), and the reference data has been cited in this work in line 389.
